

# Influence of a hyperlipidic diet on the composition of the non-membrane lipid pool of red blood cells of male and female rats

Xavier Remesar[1,2,4], Arantxa Antelo[1,4], Clàudia Llivina[1],
Emma Albà[1], Lourdes Berdié[3], Silvia Agnelli[1], Sofía Arriarán[1],
José Antonio Fernández-López[1,2,4] and Marià Alemany[1,2,4]

[1] Department of Nutrition and Food Science, Faculty of Biology, University of Barcelona, Barcelona, Spain
[2] Institute of Biomedicine, University of Barcelona, Barcelona, Spain
[3] Scientific & Technical Services, University of Barcelona, Barcelona, Spain
[4] CIBER OBN, Barcelona, Spain

Corresponding author
Marià Alemany, malemany@ub.edu

## ABSTRACT

**Background and objectives.** Red blood cells (RBC) are continuously exposed to oxidative agents, affecting their membrane lipid function. However, the amount of lipid in RBCs is higher than the lipids of the cell membrane, and includes triacylglycerols, which are no membrane components. We assumed that the extra lipids originated from lipoproteins attached to the cell surface, and we intended to analyse whether the size and composition of this lipid pool were affected by sex or diet.

**Experimental design.** Adult male and female Wistar rats were fed control or cafeteria diets. Packed blood cells and plasma lipids were extracted and analysed for fatty acids by methylation and GC-MS, taking care of not extracting membrane lipids.

**Results.** The absence of $\omega$3-PUFA in RBC extracts (but not in plasma) suggest that the lipids extracted were essentially those in the postulated lipid surface pool and not those in cell membrane. In cells' extracts, there was a marked depletion of PUFA (and, in general, of insaturation). Fatty acid patterns were similar for all groups studied, with limited effects of sex and no effects of diet in RBC (but not in plasma) fatty acids. Presence of *trans* fatty acids was small but higher in RBC lipids, and could not be justified by dietary sources.

**Conclusions.** The presence of a small layer of lipid on the RBC surface may limit oxidative damage to the cell outer structures, and help explain its role in the transport of lipophilic compounds. However, there may be other, so far uncovered, additional functions for this lipid pool.

## INTRODUCTION

Most blood-borne lipids are carried by plasma lipoproteins which play a critical role in the transfer and distribution of energy between organs and tissues. Red blood cell (RBC) integrity is essential, and largely depends on their external membranes; thus, there is abundant literature on RBC lipids, and their modulation by diet (*Gibson et al., 1984*; *Rotimi et al., 2012*; *Volek et al., 2004*) or disease (*Ferreri et al., 2005*; *Labagambe et al., 2008*; *Lemaitre et al., 2002*).

Analysis of RBC lipid classes showed the presence of different types of phospholipids, cholesterol (and its esters), but also significant amounts of triacylglycerols (TAG) (*Rotimi et al., 2012*; *Volek et al., 2004*). However, TAG are not components of cell membranes; mammal RBCs could not oxidize lipids, and thus are devoid of reserves. Consequently, the presence of TAG in RBC, widely demonstrated (*Volek et al., 2004*), remains unexplained, since mammalian RBCs do not contain intracellular membranous structures. The outstanding presence of TAG adds to the ample variability on the proportions of lipid classes (and fatty acids) found by different studies (*Carlson, Carver & House, 1986*; *Rotimi et al., 2012*) in RBC. These data, taken together, hint at the existence of lipid depots in RBCs that are different from (true) membrane lipids. As far as we know, however, in RBC no specific lipid stores have been described which could explain the presence of TAG and other lipid classes in excess of those forming part of cell membrane structure.

RBCs are in constant contact with plasma lipids (and lipophilic hormones); they have a direct relation with apolipoproteins (*Cooper, Durocher & Leslie, 1977*), bind lipoproteins (*Hui, Noel & Harmony, 1981*), and have been found to carry free cholesterol to the liver (*Hung et al., 2012*). RBCs carry/harbour a number of steroid hormones (*Romero et al., 2012*), as well as extra glucose and amino acids loosely adsorbed on their surface (*Proenza, Palou & Roca, 1994*). This disposition has been suggested to speed up the interchange of substrates with the epithelial cells lining the capillary walls (*Elwyn et al., 1972*).

RBC membranes are periodically in close contact with those of endothelial cells due to constant blood cycling. This also causes cyclic exposure of the membranes to contact with different tissues and blood changes (arterial/venous) in pH and $pO_2$. These changes compound the continuous variations in other plasma components (fatty acids, lipoproteins, cholesterol, glucose, etc.) and exposure to free radicals (superoxide, peroxynitrite, nitric oxide) in tissues (*Kagota et al., 2009*). Consequently, the structure and fatty acid composition of RBC membrane lipids change with their age, in spite of an active lipid turnover, affecting especially its external layer (*Dise, Goodman & Rasmussen, 1980*; *Quarfordt & Hilderman, 1970*), which composition is markedly different from the more stable inner layer (*Zwaal & Schroit, 1997*). In any case, constant exposure results in a relative loss of function, with increased rigidity, a condition aggravated in metabolic syndrome (*Van Blitterswijk, Van der Meer & Hilkmann, 1987*).

Studies on the incorporation of labelled fatty acids showed a small but significant incorporation of them in RBCs in just 24 h (*Leyton, Drury & Crawford, 1987*); this implies the existence of a rapid system for plasma fatty acid interchange with RBC lipids. This could not be fully explained by phospholipid or cholesterol turnover (*Quarfordt*
& *Hilderman, 1970*; *Reed, Murphy & Roberts, 1968*), which are slower and related to median RBC half-life: 45–50 days in rats (*Burwell, Brickley & Finch, 1953*). In addition, the substitution of internal membrane phospholipids is extremely slow (*Zwaal & Schroit, 1997*). We hypothesized that the rapid lipid interchange repeatedly observed in RBC may be related to a more accessible pool, in equilibrium with plasma, rather than to the necessarily stable cell membrane lipid bilayer.

A report suggest the presence of a fuzzy (probably lipid) cover on the RBC of hyperlipidemic blood when examined under the microscope (*Miller, Hirani & Bain, 2013*), and the overall lipid content of packed blood cells is higher than what can be expected from its membrane lipids alone. We hypothesized that the ability of RBC to transport lipophilic compounds and the presence of non-membrane lipids may be a consequence of the more or less loose presence of lipoprotein-derived lipids attached/bound to the surface of the RBC. This pool may participate in the turnover/repletion of RBC membrane lipid fatty acids. In order to test the hypothesis we analysed RBC lipids using a known mild lipid extraction procedure (i.e., assumedly not affecting their membrane lipids), comparing the fatty acid patterns and quantitative analysis of the lipids extracted in function of sex and exposure to a hyperlipidic diet.

## MATERIALS & METHODS

### Animals, diets, and experimental setup

All animal handling procedures were carried out in accordance with the norms of European, Spanish and Catalan Governments. The Animal Ethics Committee of the University of Barcelona approved the specific procedures used.

Nine week old female and male Wistar rats (Harlan Laboratory Models, Sant Feliu de Codines, Spain) were used. Six animals per group were housed in two-rat cages, had free access to water. The animals were kept in a controlled environment (lights on from 08:00 to 20:00; 21.5–22.5 °C; 50–60% humidity). Two groups of animals for each sex were randomly selected and were fed *ad libitum*, for 30 days, with either normal rat chow (Harlan #2014) or a simplified cafeteria diet (*Ferrer-Lorente et al., 2005*). This diet was made up by chow pellets, plain cookies, with liver pâté, bacon, whole milk with 300 g/L sucrose and a mineral plus vitamin supplemens. We used the procedures for food intake estimation and analysis described previously (*Prats et al., 1989*). Diet composition was (expressed as energy content): carbohydrate 67%, protein 20%, and lipid 13% for controls; the mean composition of the cafeteria diet ingested was carbohydrate 47%, protein 12% and lipid 41%. This diet induced a significant increase in body fat and has been used for a long time in comparative studies on metabolic syndrome (*Ferrer-Lorente et al., 2010*; *Ferrer-Lorente et al., 2005*; *Romero et al., 2009*).

The known composition and analysed fatty acid composition of the food items (including the control rat chow pellets) allowed us to estimate the energy and nutrient content of the diets consumed. Table 1 shows the amount of each food consumed per rat and day, as well as their nutrient energy equivalences of the four experimental groups ($N = 6$ for each) female-control (FC), female-cafeteria (FK), male-control (MC) and male-cafeteria (MK).

**Table 1 Food intake of control and cafeteria diet-fed rats.** The data are the mean ± sem of at least 20 days taken from three cages (2 rats in each) per group. Statistical significance of the differences between groups (2-way ANOVA): the columns represent the *P* values for each variable: sex and diet.

| | Units | Male rats | | Female rats | | *P* values | |
|---|---|---|---|---|---|---|---|
| | | Control | Cafeteria | Control | Cafeteria | Sex | Diet |
| Nutrient energy intake | | | | | | | |
| Total energy | kJ/d | 272 ± 15 | 630 ± 13 | 209 ± 13 | 527 ± 10 | <0.0001 | <0.0001 |
| Carbohydrate | kJ/d | 184 ± 11 | 312 ± 4 | 141 ± 6 | 274 ± 9 | <0.0001 | <0.0001 |
| Lipid | kJ/d | 33.3 ± 1.6 | 259 ± 4 | 25.6 ± 2.0 | 210 ± 10 | <0.0001 | <0.0001 |
| Protein | kJ/d | 54.8 ± 2.3 | 78.2 ± 1.4 | 40.1 ± 3.1 | 64.2 ± 5.0 | 0.0002 | <0.0001 |
| Food items' intake | | | | | | | |
| Rat chow | g/d | 22.5 ± 1.3 | 5.53 ± 0.45 | 17.3 ± 1.1 | 4.95 ± 0.95 | 0.0079 | <0.0001 |
| Sugared milk | mL/d | | 19.9 ± 1.1 | | 20.6 ± 0.4 | NS | <0.0001 |
| Plain cookies | g/d | | 11.7 ± 0.9 | | 8.74 ± 1.03 | 0.0384 | <0.0001 |
| Bacon | g/d | | 7.26 ± 0.55 | | 5.06 ± 0.32 | 0.0025 | <0.0001 |
| Pâté | g/d | | 6.76 ± 0.36 | | 5.56 ± 0.24 | 0.0117 | <0.0001 |

## Sampling

At the end of the experiment, the animals were anesthetized with isoflurane and immediately killed by exsanguination from the exposed aorta, using dry heparinized syringes. Blood was centrifuged 20 min at $2,000 \times g$, at 2–4 °C. Plasma was frozen; the plasma-free packed cells were also frozen; all samples were kept at −20 °C.

Packed cell volume was calculated from its weight and density (previously measured: 1.11 g/mL); the percentage ratio of this volume *vs.* that of blood gave us the haematocrit value (Hc). Under the conditions of centrifugation used, packed cells included 9.4% of its volume as trapped plasma, calculated according to previously published data (*Romero et al., 2012*).

## Plasma general analytical procedures

Plasma triacylglycerols and total cholesterol were measured using the Biosystems kits #11828, and #11505, respectively. Plasma non-esterified fatty acids were estimated with kit NEFA-HR(2) (Wako, Neuss, Germany).

## Sample lipid extraction

Samples of 0.050 mL of just thawed plasma, or about 0.20 g of frozen packed cells, were suspended (and gently vortexed) in 10 mL of trichloromethane: methanol (3:1 v/v) (*Folch, Lees & Sloane-Stanley, 1957*) in screw-cap tubes with Teflon liners. The samples were extracted for 24 h in rotary mixers at room temperature. Then 2 mL of 9 g/L NaCl in water were added, and the extraction was continued for 1 h. The aqueous supernatants, and, eventually, interface protein, were discarded. The organic phase was carried to clean tubes and dried under a gentle stream of nitrogen at room temperature; the lipid residue was used for fatty acid derivatization. This extraction procedure was repeated using samples of 0.500 ml of plasma and about 0.5 g of packed cells, but now the dry residue

was carefully weighed in order to measure the total lipid extracted from the samples using this procedure.

Food samples were powdered under liquid nitrogen and extracted overnight with trichloromethane: methanol (3:1 v/v) (*Folch, Lees & Sloane-Stanley, 1957*) and the samples processed for fatty acid analysis using the same procedure described for tissue samples.

## Fatty acid analysis

Lipid residues were used for methylation (*Christie, 1993*). In short, they were suspended in 0.50 mL of 100 g/L boron trifluoride in methanol (Fluka, Buchs, Switzerland) (i.e., 116 µmol $F_3B$), taking care to suspend and dissolve all residues. The tubes were left standing in the dark at 4 °C for 12 h, coarsely covered with aluminium foil. Later, 1 mL hexane (Panreac, Castellar del Vallès Barcelona, Spain) and 2 mL pure water were added; the mixture was vortexed, the tubes capped again and left in an orbital rotary mixer for 15 min. The upper (aqueous) phase was transferred to another tube, which was again brought to dryness under a gentle stream of nitrogen. The residues were dissolved in 0.150 mL of HPLC-quality hexane (Panreac). The whole volume was then transferred to 0.200 mL Mandrel GLS inserts (BC Scientific, Miami, FL, USA) within Agilent screw cap vials (Agilent, Santa Clara, CA, USA) which had 8 mm PTFE/silicone septa (Soltec, Bether, CT, USA). The samples were kept tightly closed at −20 °C until measurement.

Samples were analysed with a CG-MS system (QP2010; Shimadzu, Kyoto, Japan) using a SP-2560 Supelco (Supelco, Bellefonte, PA, USA) column. The samples were run using, as standards, an extended methylated fatty acid mixture (Supelco FAME mix C4–C24). Calculations were done using the Shimadzu FASST for GC-MS program (version 2).

The rates of recovery of lipids (and in particular fatty acids) were analysed with internal standards of bis-C17:0 diacylglycerol (Sigma) randomly added to a number of duplicate samples.

## Calculations

The contribution of trapped plasma in packed blood cells to total lipids and to each individual FA measurement was calculated for each individual rat from their matching analyses of plasma and RBC-extracted lipids (Table 2).

The approximate amount of RBC membrane lipid was estimated from the mean rat cell volume, 69 µm$^3$ (69 fL) (*Balazs, Grice & Airth, 1960*), normal cell counts ($7.2 \times 10^6$ cells/µL of blood) (*Balazs, Grice & Airth, 1960*), corrected by the hematocrit value. The mean cell diameter (6.7 µm) was that of the biconcaval RBC flattened disk. If we calculate the diameter of a sphere with the same volume than the actual RBC, we would obtain a smaller diameter, 5.08 µm, but the actual surface area of the RBC is higher than that of a sphere of the same volume. By comparing data on human RBCs (i.e., 90 fL volume were equivalent to a surface of about 136 µm$^2$ (*McLaren, Brittenham & Hasselblad, 1987*) we obtained diameters for volume- or surface-equivalent spheres of, respectively, 5.5 and 6.6 µm. That is, the real RBC surface was equivalent to that of a sphere with a volume about 20% higher than that obtained from the simple translation of the actual RBC volume to a sphere. Applying the same relationship to rat cells, the "sphere diameter" was

**Table 2  Blood lipid distribution in female and male rats subjected 30 days to a cafeteria diet.** All values are the mean ± sem of 6 different animals per group. Statistical analysis was done using a 2-way ANOVA.

| | Units | Male rats | | Female rats | | P values | |
| --- | --- | --- | --- | --- | --- | --- | --- |
| | | Control | Cafeteria | Control | Cafeteria | Sex | Diet |
| Plasma triacylglycerols | mM | 1.28 ± 0.07 | 1.40 ± 0.11 | 1.35 ± 0.08 | 1.48 ± 0.07 | NS | NS |
| Plasma total cholesterol | mM | 1.76 ± 0.11 | 2.15 ± 0.17 | 2.25 ± 0.21 | 2.56 ± 0.13 | 0.0404 | 0.0106 |
| Plasma free fatty acids | mM | 0.35 ± 0.04 | 0.43 ± 0.03 | 0.75 ± 0.04 | 0.68 ± 0.08 | <0.0001 | NS |
| Blood cell extracted lipid | mg/g[a] | 12.0 ± 3.5 | 11.5 ± 2.1 | 12.8 ± 3.2 | 10.1 ± 2.8 | NS | NS |
| FA extracted from RBC | μmol/g[a] | 21.9 ± 0.4 | 20.8 ± 6.6 | 23.2 ± 1.9 | 18.2 ± 2.6 | NS | NS |
| FA in trapped plasma | μmol/g[a] | 0.37 ± 0.04 | 0.88 ± 0.29 | 0.68 ± 0.08 | 0.82 ± 0.30 | NS | NS |
| | %[b] | 1.7 ± 0.2 | 3.6 ± 1.6 | 3.0 ± 0.4 | 5.5 ± 1.8 | NS | NS |
| Packed RBC volume | % | 41.3 ± 0.6 | 44.1 ± 1.6 | 40.9 ± 0.9 | 42.1 ± 0.6 | NS | NS |

**Notes.**

FA, fatty acids.

[a] Data per gram of fresh packed cells.

[b] Percentage (in weight) of fatty acids in trapped plasma with respect to total fatty acids recovered from cells.

increased 20% to obtain an estimation of RBC surface; this way, we obtained a probably better approximation to a sphere with the surface area of a rat RBC using a mean diameter of 6.1 μm. The corresponding mean individual surface area of a rat RBC would then be 117 μm$^2$. Since the thickness of a RBC bilayer membrane is in the range of 8–10 nm (*Shkulipa, 2006*), we used a mean value of 9 nm. These data allowed an estimation of the total volume of membrane (i.e., lipid bilayer) in a single RBC: surface area multiplied by the layer thickness, i.e., $117 \times 0.009 = 1.05$ μm$^3$. Since the corrected haematocrit value was in the range of 43%, in 1 ml of packed cells there will be about $16.7 \times 10^9$ cells, and thus the lipid bilayer volume in 1 ml of packed cells will be: $1.05 \times 16.7 \times 10^9$ μm$^3$, i.e., 17.5 μL. The density of lecithin (as representative membrane lipid) is 1.03 g/mL, thus, the weight of lipids in 1 ml (i.e., 1.1 g) of packed RBC would be in the range of 18.4 mg (1.7% w/w). This accounts for about half of the lipids estimated in packed RBC, a value concordant with the 3% of lipid contained in clotted animal blood, when analysed as food.

The normal composition of RBC phospholipid, a main lipid component (*Vayá et al., 1993*), is known (*Cooper, Durocher & Leslie, 1977*; *Pöschl et al., 1999*); consequently, the weight of fatty acids account for about 57% of the total membrane lipid. Taking oleic acid a "model" for molecular weight and abundance (an oleoyl residue has a molecular weight of 270), and applying this value and the proportion of fatty acids to the estimated weight of membrane lipid for packed RBC we obtain about 39 μmol fatty acids per g of packed cells. This figure is an approximate estimate of the amount of lipids expected in packed RBC if the bilayer membrane was the only source of cell lipid.

## Statistical methods

Statistical analyses were carried out with two- or three-way ANOVA comparisons, using the Statgraphics Centurion XVI program package (Statpoint Technologies, Warrengton, VA, USA).

**Table 3 Sum of SFA, MUFA, PUFA and trans-FA levels in plasma and blood lipids of male and female rats subjected to a control or a cafeteria diet for 30 days.** All values are the mean ± sem of 6 different animals per group. Statistical analysis was done using a 2-way ANOVA.

|  | Units | Male rats | | Female rats | | P values | |
|---|---|---|---|---|---|---|---|
|  |  | Control | Cafeteria | Control | Cafeteria | Sex | Ciet |
| **Plasma** |  |  |  |  |  |  |  |
| SFA | mM | 1.53 ± 0.17 | 3.25 ± 1.02 | 2.16 ± 0.25 | 3.67 ± 1.18 | 0.0300 | NS |
| MUFA | mM | 0.76 ± 0.12 | 2.42 ± 0.91 | 0.96 ± 0.10 | 2.05 ± 0.88 | NS | 0.0235 |
| $\omega$-3 PUFA | mM | 0.027 ± 0.005 | 0.038 ± 0.014 | 0.027 ± 0.005 | 0.032 ± 0.014 | NS | NS |
| $\omega$-6 PUFA | mM | 1.75 ± 0.24 | 4.12 ± 1.35 | 2.11 ± 0.31 | 3.43 ± 1.32 | NS | 0.0435 |
| *trans*-FA | mM | 0.007 ± 0.004 | 0.006 ± 0.004 | 0.005 ± 0.001 | 0.014 ± 0.008 | NS | NS |
| **RBC** |  |  |  |  |  |  |  |
| SFA | µmol/g | 12.43 ± 0.77 | 11.52 ± 3.22 | 13.54 ± 1.25 | 9.52 ± 0.61 | NS | NS |
| MUFA | µmol/g | 6.78 ± 0.47 | 6.60 ± 2.49 | 6,62 ± 0.79 | 6.04 ± 0.83 | NS | NS |
| $\omega$-3 PUFA | µmol/g | 0.00 | 0.00 | 0.00 | 0.00 |  |  |
| $\omega - 6$ PUFA | µmol/g | 2.58 ± 0.08 | 2.59 ± 1.25 | 2.92 ± 0.28 | 2.50 ± 1.08 | NS | NS |
| *trans*-FA | µmol/g | 0.12 ± 0.02 | 0.11 ± 0.03 | 0.08 ± 0.03 | 0.12 ± 0.01 | NS | NS |

**Notes.**

FA, fatty acids.

## RESULTS

### Plasma lipids

There were no significant differences in plasma triacylglycerols and packed cell volume between the four groups of rats studied (Table 2). However, total cholesterol was modified by sex and diet (higher for females and cafeteria diet). Non-esterified fatty acids were unaffected by diet, but females showed higher plasma levels than males. There were no clear relationships between the plasma lipid parameters. Table 2 also shows the proportion of lipids recovered from packed RBC. There were no differences attributable to sex and diet, the data was remarkably uniform.

The patterns of distribution of individual fatty acids in plasma total-lipids were also similar between the sex/diet groups (Fig. 1). There were only a few statistical differences between individual fatty acids between them. In plasma, exposure to a cafeteria diet resulted in significant differences in palmitoleic and oleic acids. Sex affected the levels of stearic, $\alpha$-linolenic, eicosadienoic, gondoic and heneicosanoic acids. The similitude of overall distribution pattern was repeated when fatty acids were grouped in their main classes: saturated (SFA), monounsaturated (MUFA), polyunsaturated (PUFA, $\omega$-3 and $\omega$-6) and *trans* (Table 3); SFA and PUFA (mostly $\omega$-6) were predominant, there were only trace amounts of $\omega$-3 PUFA and *trans* fatty acids were practically absent. Only a few female rats treated with the cafeteria diet showed measurable levels of *trans* fatty acids. However, the grouped sums of fatty acids showed a significant effect of diet for MUFA and $\omega$-6 PUFA, and effects for sex in SAT fatty acids. Total plasma fatty acids (i.e., the sum of all individual fatty acids analysed, expressed as mM) were 4.1 ± 0.4 (male control), 9.8 ± 3.3 (male cafeteria), 5.3 ± 0.6 (female control), and 9.2 ± 3.3 (female cafeteria); the differences were significant ($P = 0.029$) for diet but not for sex.

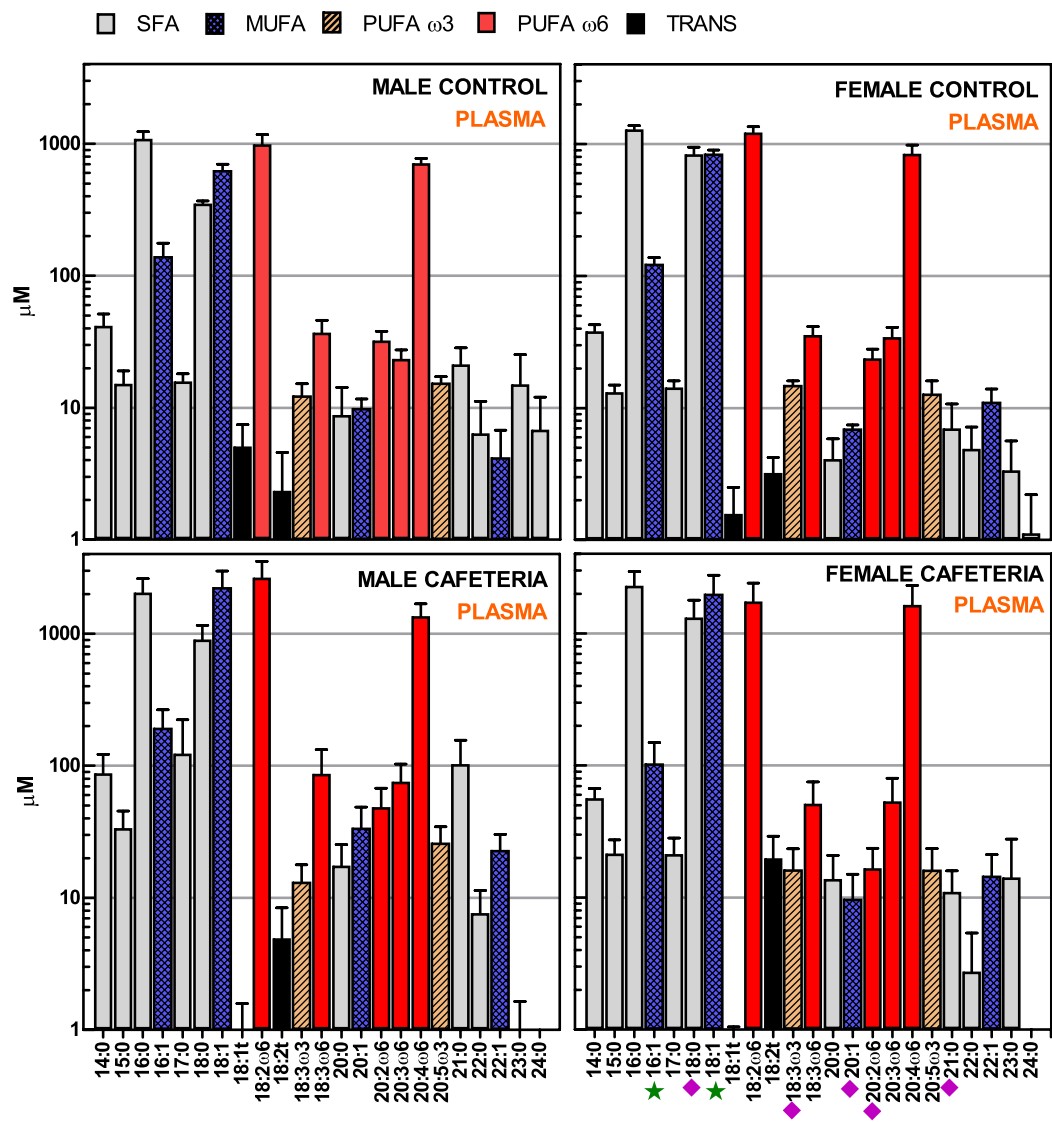

**Figure 1** **Fatty acids present in plasma lipids of male and female rats, after exposure of 30 days to a hyperlipidic cafeteria diet, compared with controls.** The data are the mean ± sem of 6 different animals per group. Pale grey, SFA (saturated fatty acids); cross-patterned blue, MUFA (monounsaturated fatty acids); dash-patterned orange, ω-3 PUFA (polyunsaturated fatty acids); red, ω-6 PUFA; black, *trans* fatty acids. Statistical significance of the differences between groups (2-way ANOVA): green star, $P < 0.05$ for diet; purple diamond, $P < 0.05$ for sex.

## Red blood cell extractable lipids

Figure 2 shows the individual fatty acid levels in packed cell extracts of the four groups of rats. The patterns were highly similar, with only small differences induced by diet (palmitic and heneicosanoic acids), and none due to the effect of sex. In addition, there was a lesser variety of fatty acids than in plasma, with lower levels of PUFAs, and a small but clear presence of *trans* fatty acids (essentially elaidic acid). When considering the classes of fatty acids (Table 3), all groups showed a similar pattern, with low ω-6 PUFA, nil presence of ω-3 PUFA and a token presence of *trans* fatty acids.

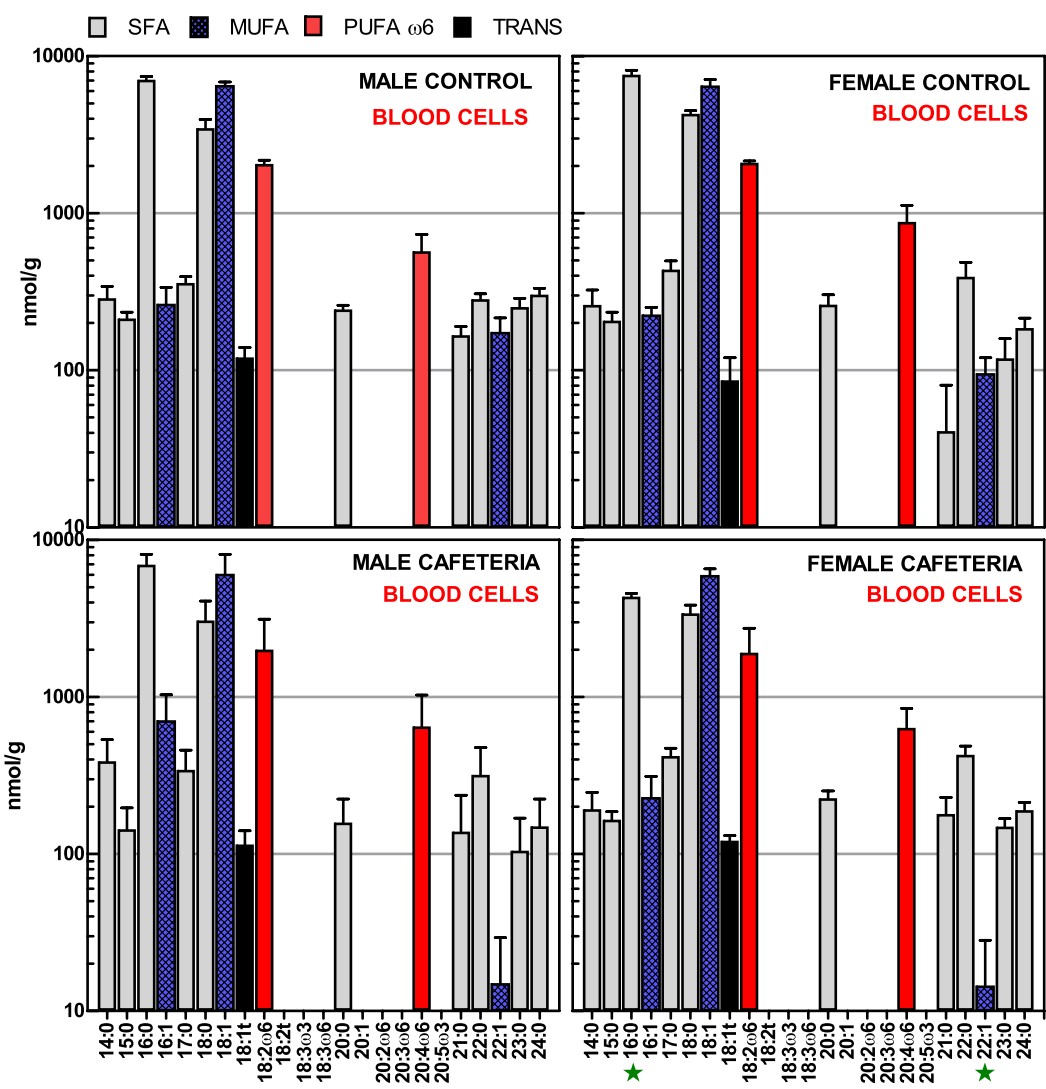

**Figure 2 Fatty acids present in packed blood cell lipids of male and female rats, after exposure of 30 days to a hyperlipidic cafeteria diet, compared with controls.** The data are the mean ± sem of 6 different animals per group, and are expressed in nmol/g of fresh cells. Pale grey, SFA; cross-patterned blue, MUFA; red, $\omega$-6 PUFA; black: *trans* fatty acids. Statistical significance of the differences between groups (2-way ANOVA): green star, $P < 0.05$ for diet.

## Double bond distribution

The differences between plasma and RBC fatty acids widened when the number of double bonds was computed. The lower proportion of PUFA in RBC compared with plasma, and the nil influence on this parameter of SFA, resulted in a proportion of double bonds in the range of 0.5 per fatty acid molecule in RBC versus almost 1.5 in plasma lipids (Fig. 3). The proportion of double bonds in PUFA with respect to the total sum of double bonds showed the same pattern; in plasma, insaturation was mostly due to PUFA, but in RBC, the share of MUFA was much higher. Finally, the proportion of *trans* double bonds with respect to the sum of total double bonds was small but significantly higher in the RBC lipids of all

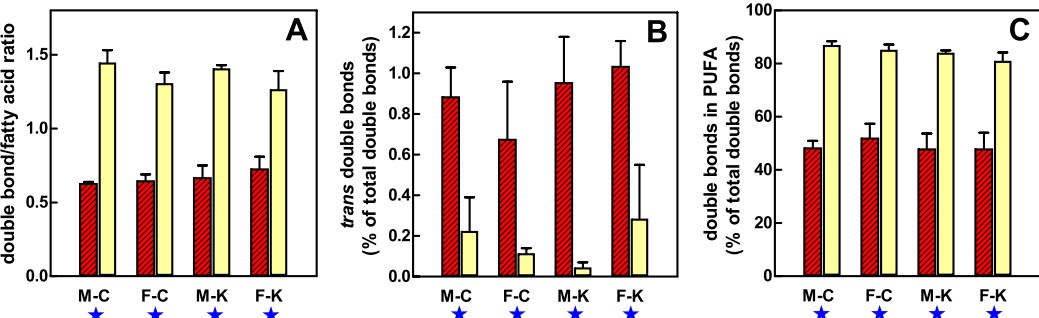

**Figure 3** **Distribution of double bonds in the lipids of RBC and plasma of male and female rats, during exposure of 30 days to a hyperlipidic cafeteria diet, compared with controls.** (A) presents the mean number of double bonds per fatty acid molecule in RBC lipids (dashed red columns) and plasma (pale yellow columns). (B) depicts the percentage of trans fatty acid double bonds with respect to the total double bonds in RBC and plasma lipids. (C) shows the percentage of double bonds that correspond to combined PUFAs in RBC and plasma lipids. All data are the mean ±sem of 6 animals per group. M-C, male control; F-C, female control; M-K, male cafeteria, F-K, female cafeteria. Statistical significance of the differences between compartments (cell vs. plasma lipids) calculated using a three-way ANOVA: a blue star, $P < 0.05$. There were no significant differences related to sex or diet.

experimental groups than in plasma, where *trans* fatty acids showed higher variability and lower values.

### Estimation of fatty acids intake

Table 4 presents an estimation of the mean intake (in mmol/day) of dietary fatty acids for controls and cafeteria-fed rats. The data were calculated from the consumption of each type of food item per cage and day and the composition in fatty acids of the foods offered to the rats shown in Table 1.

The distribution of the different types of fatty acids in both standard chow and self-selected cafeteria diet were considerably different, both in proportions and (in part) variety, giving rise to widely marked differences for sex, significant for all fatty acids except for the short-chain capric, lauric and myristic acids. The cafeteria diet-fed rats ingested daily a much higher proportion (and variety) of fatty acids than controls ($P < 0.0001$ for all fatty acids studied), and male rats ate more than females.

An analysis of the correlation between daily fatty acid intake *vs.* circulating plasma fatty acids showed that the only correlations observed (using all animals) were for oleic and gondoic acids ($P = 0.026$ and $P = 0.046$, respectively), there were no correlations for any of the other fatty acids, and the significance was lost when analysing the four groups of animals separately.

### DISCUSSION

A critical aspect of the validity of the data presented here showing the existence of a secondary lipid pool in RBC, different from that constituted by membrane lipids, is the comparison of the methodology used and the fatty acid profiles. Those found were different from those expected from membrane composition as described by other authors (*Cooper, Durocher & Leslie, 1977*; *Ferreri et al., 2005*; *Lemaitre et al., 2002*).

**Table 4 Fatty acid intake of control and cafeteria-fed diet rats.** The data were calculated from the mean fatty acid composition of all foods given to the rats and the data of consumption indicated in Table 1. Values are the mean ± sem. Statistical significance of the differences between groups (2-way ANOVA): the columns represent the $P$ values for each variable: sex and diet.

| Fatty acid mmol/day | | Male rats | | Female rats | | $P$ values | |
|---|---|---|---|---|---|---|---|
| | | Control | Cafeteria | Control | Cafeteria | Sex | Diet |
| Capric | 10:0 | <0.1 | 0.52 ± 0.03 | <0.1 | 0.43 ± 0.04 | NS | <0.0001 |
| Lauric | 12:0 | <0.1 | 2.66 ± 0.22 | <0.1 | 2.20 ± 0.21 | NS | <0.0001 |
| Myristic | 14:0 | 0.03 ± 0.0 | 1.12 ± 0.07 | 0.02 ± 0.0 | 0.93 ± 0.07 | NS | <0.0001 |
| Palmitic | 16:0 | 0.44 ± 0.02 | 5.32 ± 0.21 | 0.34 ± 0.01 | 4.17 ± 0.13 | <0.0001 | <0.0001 |
| Stearic | 18:0 | 0.08 ± 0.00 | 1.94 ± 0.09 | 0.06 ± 0.00 | 1.54 ± 0.06 | 0.0009 | <0.0001 |
| Arachic | 20:0 | 0.12 ± 0.00 | 0.23 ± 0.00 | 0.09 ± 0.00 | 0.18 ± 0.00 | <0.0001 | <0.0001 |
| Behenic | 22:0 | 0.16 ± 0.00 | 0.35 ± 0.00 | 0.13 ± 0.00 | 0.27 ± 0.00 | <0.0001 | <0.0001 |
| Lignoceric | 24:0 | <0.1 | 0.43 ± 0.02 | <0.1 | 0.34 ± 0.01 | <0.0001 | <0.0001 |
| Palmitoleic | 16:1 | <0.1 | 0.42 ± 0.02 | <0.1 | 0.32 ± 0.02 | <0.0001 | <0.0001 |
| Oleic | 18:1 | 0.56 ± 0.02 | 6.99 ± 0.27 | 0.42 ± 0.02 | 3.38 ± 0.16 | <0.0001 | <0.0001 |
| Elaidic | *trans*18:1 | <0.1 | <0.1 | <0.1 | <0.1 | – | – |
| Gondoic | 20:1 | 0.11 ± 0.0 | 0.25 ± 0.0 | 0.09 ± 0.0 | 0.19 ± 0.0 | <0.0001 | <0.0001 |
| Erucic | 22:1 | <0.1 | 0.15 ± 0.1 | <0.1 | 0.12 ± 0.01 | <0.0001 | <0.0001 |
| Linoleic | 18:2 | 1.61 ± 0.06 | 2.15 ± 0.04 | 1.24 ± 0.05 | 1.63 ± 0.02 | <0.0001 | <0.0001 |
| Linolenic | 18:3 | 0.08 ± 0.00 | 0.20 ± 0.00 | 0.06 ± 0.00 | 0.15 ± 0.00 | <0.0001 | <0.0001 |
| Arachidonic | 20:4 | <0.1 | 0.14 ± 0.01 | <0.1 | 0.11 ± 0.00 | <0.0001 | <0.0001 |

The usual procedure for the estimation of blood cell-membrane fatty acids, followed almost universally is: (A) Separation and washing of RBC in order to remove all traces of plasma lipids. (B) Breakup of the cells and separation of a membrane fraction clean of non-membrane proteins, especially haemoglobin, but also structural fibres such as spectrin. (C) Solvent extraction (twice in most cases) with a suitable solvent and extraction conditions and time (*Rose & Oklander, 1965*), the remaining lipid (bound to proteins) being usually discarded. (D) Overall methylation, usually in strong acid to hydrolyse complex phospholipids. (E) Analysis through GC or CG-MS. (F) Presentation of the results as percentages of total fatty acids measured, since step (B), and in part step (C), could hardly be quantitative, and the effectiveness of step (D) is often problematic.

Since we assumed that the most probable place for the RBC non-membrane-bilayer lipid "stores" (i.e., the site containing the TAG found in RBC lipid extracts) was the cell surface, we decided not to wash the cells and extrude the plasma by centrifugation (but taking care to limit cell breakage). We estimated the mass of trapped plasma, which allowed discounting its contribution to the total (mildly) extracted lipids of packed cells. Evidently, our intention was to keep the membranes as inaccessible as possible to our extraction method, and thus we used a procedure suitable for the extraction of all lipid classes (*Folch, Lees & Sloane-Stanley, 1957*) which has been found inadequate/ ineffective for RBC membrane lipid extraction (*Eder, Reichlmayr-Lais & Kirchgeßner, 1993*; *Rose & Oklander, 1965*).

The extracted lipids were methylated with a method sufficiently powerful to release methyl-fatty acids from most phospholipids (*Christie, 1993*), but which, in turn, was not sufficiently strong for all classes of complex membrane phospholipids (*Eder, 1995*); in addition, we used internal standards to check the effectiveness of methylation.

The results obtained suggest that the RBC lipid extract was not representative of membrane lipids. First because of the practical absence of critical PUFA membrane components (such as docosahexaenoic), even when compared with plasma. Second because of the impoverishment in double bonds, compared with plasma (which we assume was its origin), essentially at the expense of PUFA. There was, also, a relative abundance of SFA. Both factors reflected a situation extremely different from that found in membranes (*Carlson, Carver & House, 1986*; *Lemaitre et al., 2002*). In any case, these results do not preclude the possibility that an unknown proportion of RBC membrane lipids would be extracted with the procedure used. The trichloromethane: methanol method indeed can dissolve most lipids from complex matrices, including membranes, as in brain (*Folch et al., 1951*), but we did not break massively all cells, only compressed and snap froze them, preserving in part their structure. In addition, we discounted the lipids of trapped plasma. Nevertheless, since the mass of lipid recovered from cells was similar to the calculated weight of membrane lipids, a higher share of PUFA should be detected if membrane lipids were extracted in a significant proportion. Contrary to that, the results obtained strongly suggest that membrane lipids were not extracted in a proportion high enough to allow us to detect their typical PUFA presence/pattern.

Most studies on RBC lipids first purified cell membranes from washed cells' ghosts (*Eder, Reichlmayr-Lais & Kirchgeßner, 1993*; *Rose & Oklander, 1965*; *Vayá et al., 1993*). Using these methods, PUFA levels were higher, in contrast with studies analyzing "erythrocyte lipids" (*Labagambe et al., 2008*; *Rotimi et al., 2012*; *Volek et al., 2004*). The differences between these approaches support our assumption that the RBC lipid we obtained represented largely non-membrane RBC lipids.

The quantitative importance of the extracted lipids was highly dependent on diet, but not on sex, with a proportion of 0.26 to 0.89% w/wet RBC weight, i.e., 1–2% w/dry RBC weight. However, the generalized lack of correlation of dietary fatty acid intake and the levels found in plasma lipids showed that these relationships are not straightforward and may be modulated by overall energy (largely lipid/ glucose) metabolism and substrate plasma turnover.

The easy availability for extraction of these RBC lipids, together with their abundance in MUFA and SFA agree with a lipoprotein origin of the deposits and their placement on the RBC surface. The findings of fuzzy borders in direct microscopic examination of blood cells from hyperlipidic plasma (*Cooper et al., 1975*; *Miller, Hirani & Bain, 2013*) may support this assumption. In addition, as far as we know, no internal RBC membrane or lipid depots have been described for mammals. However, the lack of significant effects of diet (hyperlipidic in our case) on the mass of recovered RBC lipids suggests that the lipid "cover" of RBC should be rather thin (it would be similar in thickness to the membrane lipid bilayer if it were spread uniformly) and not directly dependent on

plasma lipid content. The lack of relationship seems to preclude the occasional weak bonding of lipoproteins as such, a possibility based on the presence of anchor proteins for apolipoproteins on the RBC surface (*Hui, Noel & Harmony, 1981*). The thin layer (or discontinuous blobs) of lipid is bound, probably, rather permanently to the actual membrane, since its proportion of double bonds (compared with plasma), essentially from PUFA, and the small presence of *trans*fatty acids suggests a sustained exposure of these lipids to oxidizing or nitrating agents (at least to a higher degree than their plasma counterparts). In any case, these differences show that their turnover is slower than that of plasma lipids.

Nitric oxide is known to favour the conversion of *cis* to *trans* double bonds (*Proell et al., 2002*), lowering the fluidity of membranes and affecting their function. Oxidative attacks by superoxide and other free radicals tend to break down unsaturated fatty acids, mainly PUFA (*Mattson & Grundy, 1985*; *Trotschansky & Rubbo, 2008*), which levels tend to decrease in structures continuously exposed to oxidizing environments. This is in part corrected by turnover of membranes in cells, and by interchange with lipoproteins in RBC (*Cooper et al., 1975*; *Dise, Goodman & Rasmussen, 1980*; *Quarfordt & Hilderman, 1970*; *Reed, Murphy & Roberts, 1968*). However, the marked lack of PUFA (and absence of ω-3) indicate that: (A) The postulated outer layer of lipid in RBC should be rather permanent, at least enough to show the effects of oxidation and nitration on its fatty acids. (B) This lipid is repeatedly exposed (for all its functional life) to highly oxidative microenvironments in capillary beds. (C) The lipid occupies a limited and defined space on the cells, which is not directly affected by the availability (or turnover) of lipids in plasma.

In any case, there must be a certain degree of interchange of lipids between the RBC outer lipid layer and plasma lipids since its comparison with lipoprotein fatty acid patterns shows a considerable degree of similitude if PUFA are excluded. Labelled fatty acids are rapidly incorporated into RBC (*Leyton, Drury & Crawford, 1987*), and interchange or reposition of PUFA in the outer layer of RBC membranes has been previously described (*Dise, Goodman & Rasmussen, 1980*; *Reed, Murphy & Roberts, 1968*). Furthermore, diets high in PUFA decrease the stiffness of RBC membranes in metabolic syndrome (*Katan et al., 1997*; *Pöschl et al., 1999*). Probably there is a direct relationship between these phenomena, and this can be a function, so far not defined, of the external lipid layer of RBC. We postulate that it may act as an intermediate step for repairs (or protection) of the RBC membrane, since in mammals most maintenance systems must be external to the RBC, because they lack nuclei, ribosomes and most of the cell turnover machinery.

We expected, at least in the rats with overweight, that as a consequence of the cyclic exposure and close contact of RBC with endothelia there would be a marked increase in *trans* fatty acids (*Alemany, 2012*), a consequence of the higher production of nitric oxide and other oxidative and nitrating agents (*Ghasemi, Zahediasl & Azizi, 2012*). The levels of *trans* fatty acids we actually found were small, but could not be justified by the residual levels found in the diet ($<0.1$ nmol/day). Endogenous production of *trans* fatty acids is linked to the production of nitric oxide by RBC themselves or by the neighbouring endothelial lining (*Zambonin et al., 2006*). In any case, PUFA are easily affected by oxidative

and nitrative processes (*Trotschansky & Rubbo, 2008*), inducing membrane damages (*Van Blitterswijk, Van der Meer & Hilkmann, 1987*). Perhaps, the presence of the outer RBC lipid layer may help the transfer (or interchange) of *trans* or damaged fatty acids (in exchange for "fresh" PUFA) to plasma lipid for disposal elsewhere, helping to extend the functionality of RBCs for a longer time.

In the end, the presence of this lipid pool results in a net loss of PUFA, probably a consequence of oxidative processes affecting first the lipid layer over the RBC, protecting the underlying membrane. The external lipid layer could, then, constitute a first line of defence against deleterious oxidative processes that shorten the lifespan and functionality of RBCs. In metabolic syndrome, RBC half-life is reduced, probably because of increased fragility and loss of flexibility (*Kung, Tseng & Wang, 2009*). Increased dietary supply of PUFA tends to reduce the extent of this damage (*Pöschl et al., 1999*), probably via lipid interchange with lipoproteins or cells (blood, endothelial, foamy, etc.) and the external lipid pool of RBCs.

## CONCLUSIONS

We postulate the presence of a small lipid pool on the RBC surface. This layer may help minimize the effects of oxidative damage on RBC membranes, which affects the functionality and lifespan of RBCs, as shown by its marked deficit of PUFA. We speculate that the loss of PUFA is probably compensated through interchange with lipoproteins. Only residual *trans* fatty acids remained. The external lipid pool may, also help explain the role of RBCs in the transport of lipophilic compounds. The possible importance of the external lipid RBC layer should be analysed under the light of available information on the role of blood cells under conditions of hyperlipidemia and inflammation. So far, these questions have not been studied in depth, in spite of their potential importance in the transport of lipids and regulatory agents between organs and in the cell-to-cell interactions (transfer, signalling) within the tight space of tissue capillary lumen and its cell lining.

## ACKNOWLEDGEMENTS

At the time this investigation took place, C. Llivina was a postgraduate student and E. Albà an undergraduate student.

### Funding

This study was done with the partial support of grants of the Plan Nacional de Investigación en Biomedicina (SAF2009-11739, SAF2012-34895) and the Plan Nacional de Ciencia y Tecnología de los Alimentos (AGL-2011-23635) of the Government of Spain. There was also a general contribution of the CIBER-OBN to the project. S Agnelli was the recipient of a Leonardo da Vinci fellowship, and S Arriarán held a predoctoral fellowship from the Catalan Government, in both cases covering part of the time invested in this study. The funders had no role in study design, data collection and analysis, decision to publish, or preparation of the manuscript.

## Grant Disclosures

The following grant information was disclosed by the authors:

Plan Nacional de Investigación en Biomedicina: SAF2009-11739, SAF2012-34895.

Plan Nacional de Ciencia y Tecnología de los Alimentos: AGL-2011-23635.

## Competing Interests

The authors declare there are no competing interests.

## Author Contributions

- Xavier Remesar conceived and designed the experiments, analyzed the data, contributed reagents/materials/analysis tools, prepared figures and/or tables, reviewed drafts of the paper.
- Arantxa Antelo performed the experiments, analyzed the data.
- Clàudia Llivina and Emma Albà performed the experiments.
- Lourdes Berdié performed the experiments, contributed reagents/materials/analysis tools.
- Silvia Agnelli and Sofía Arriarán performed the experiments, took care of the animals, obtained samples, made additional analyses.
- José Antonio Fernández-López conceived and designed the experiments, analyzed the data, prepared figures and/or tables, reviewed drafts of the paper.
- Marià Alemany conceived and designed the experiments, contributed reagents/materials/analysis tools, wrote the paper, prepared figures and/or tables, reviewed drafts of the paper.

## Animal Ethics

The following information was supplied relating to ethical approvals (i.e., approving body and any reference numbers):

The Animal Ethics Committee of the University of Barcelona approved all the animal handling procedures applied in this study.

## Data Deposition

The following information was supplied regarding the deposition of related data:

http://hdl.handle.net/2445/66010.

## Supplemental Information

Supplemental information for this article can be found online at http://dx.doi.org/10.7717/peerj.1083#supplemental-information.

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
