# Peer review of "Influence of a hyperlipidic diet on the composition of the non-membrane lipid pool of red blood cells of male and female rats"

_PeerJ, doi:10.7717/peerj.1083_

## Round 0.1 · original submission · Major Revisions

· Academic Editor

Major Revisions

I carefully read your comments rearding your previous submission to PLOS ONE. However, I decided not to ask you for those documents and to deal with this submission as a fresh one.

You will find below the comments from the reviewers. As usual, you are not required to include all those modifications in the manuscript, but all their comments should be properly replied.

I read the mansucript myself and my main concern is that the potential role of this lipid pool in RBC as a first barrier against oxidative stress should be better explained, also in the Abstract.

Reviewer 1 ·

Basic reporting

The article is original and well written.
It includes sufficient introduction and background to demonstrate how the work fits into the broader field of knowledge.

Experimental design

In the experimental design, you mentioned a “simplified cafeteria diet” (l. 85). You should briefly explain your choice of only 4 food items. Do you think that sugared milk, cookies, bacon and pâté are sufficient to influence the lipid pool of RBC? You should better justify your choice because the components of the cafeteria diet are particularly important in your study.

In “plasma general analytical procedures” (l. 101), you described blood glucose measurement, but I did not see results on glucose…
Then, you wrote (l. 104) that triacylglycerols, total cholesterol and non-esterified fatty acids were measured using 2 different kits. Kits 11828 and 11505 respectively measured TG and CH. However, you should better explain the quantification of non-esterified fatty acids.

Validity of the findings

Results and discussion parts are coherent. However, I have some comments:

I think that your discussion about males and females fatty acids consumption (l. 227-229 – “This may be in part … (Rafecas et al. 1994)”) is not appropriate in this part of the manuscript. You should comment it in the discussion part, not in the results part.

In the figures and tables, you should ameliorate some things:
- In fig. 1: Two different symbols would be easier to differentiate than stars of different colour (green or purple).
- In fig. 2 legend: please finish your sentence “the data are mean… nmol/g of fresh”
- In fig. 3 legend: please correct “F-C” by “F-K” for female cafeteria group.
- Table 2 legend: I think sentences are missing concerning data, 1 and 2 meaning and abbreviation meaning.
- Table 3 legend: Please rewrite your first sentence “Sum of…30 days” in order to add “trans FA” and to change “blood lipids” by “RBC”. Please explain umol/g. What “NA” means? (NS?). Abbreviation meaning is also missing here.
In Table 3, you should add statistical results for RBC.

Additional comments

In your article, you discussed the influence of diet and sex on the composition of lipid pool. However, in your title you did not include the sex influence. You should add this point in the title, maybe reformulating it.

Reviewer 2 ·

Basic reporting

No comments

Experimental design

No comments

Validity of the findings

The findings are interesting but somehow still puzzling. Under the present format and the experiment performed, there is no direct evidence that their postulated surface RBC lipid layer is real. Although transport of lipophilic compounds may be explained by the presence of an external lipid layer, different other mechanism could exist.

Additional comments

The manuscript describes the lipid profile of non-membrane red blood cells (RBC) in male and female rats and their variation in response to a cafeteria diet. While the manuscript postulate an interesting point of view, about the possible presence of a small layer of lipids on the RBC surface, their biochemical determination does not directly demonstrate that this may be the case. My major concerns are outlined below.
1. Even when the authors were very careful to avoid lipid extraction from plasma membrane of RBC, the absence of omega-3-PUFA does not necessarily mean that no lipids were extracted from RBC plasma membrane. Other explanation such as low extraction affinity for omega-3-PUFA or extraction below the limit of quantitation may account for these findings.
2. Why does the author use unwashed packed blood cells? Even though the authors discount the trapped plasma volume contribution, there might be an additional artefact due to this small plasma volume.
3. The marked deficit of PUFA in their postulated surface RBC lipid layer does not necessarily mean that this might constitute a firs line of defence against oxidative stress. In fact, if this is real, they can only be a source of lipids for something else.
4. Is there any possibility that the extraction procedure extract other lipids within the RBC (not affecting the viability of the cell)? Or just a minor amount of RBC that dies and release its lipid content (and also destroyed plasma membrane containing easily extractable lipids) to the extraction solvents?
5. Triglycerides (TG) finding in their postulated surface RBC lipid layer extraction is a really puzzling finding, as plasma membrane normally do not have TG. Or either they are from plasma origin (contamination) or cell origin after appropriate biochemical esterification of fatty acids.
6. Do the authors have any idea of how a lipid pool can be maintained outside the cell? Changes in lipids between this novel pool and the plasma are direct? Or it goes through RBC?
7. The use of lipid depletion, rather than lipid enrichment (cafeteria diet) would have been a better approach for the confirmation of the postulated surface RBC lipid layer. This kind of discovery usually needs very restricted conditions. For instance RBC can participate in reverse cholesterol transport, so they can transport free cholesterol to the liver, but their function can only be determined when low HDL levels are presented with or without anaemia conditions (Arterioscler Thromb Vasc Biol. 2012;32:1460-1465). Under normal HDL levels, no effects are observed.
Based on the author findings, this reviewer do believe that there might be a small possibility that exist this postulated surface RBC lipid layer or another cellular source of lipid storage. However, under the submitted work experiments, it is just one possibility like any other, rather than a certainty. Please correct the tone through all the manuscript, particularly the abstract.

---

## Round 0.2 · accepted · Accept

· Academic Editor

Accept

I think you have properly answered all the aspects raised by the reviewers. The manuscript has improved with the modifications included and it is now acceptable for publication.

Regarding the previous submission to PLOS ONE, I agree with you that it is not necessary to include this previous history of the manuscript.